# Clinicopathological Relevance of PAX8 Expression Patterns in Acute Kidney Injury and Chronic Kidney Diseases

**DOI:** 10.3390/diagnostics12092036

**Published:** 2022-08-23

**Authors:** Maja Zivotic, Dusko Dundjerovic, Radomir Naumovic, Sanjin Kovacevic, Milan Ivanov, Danijela Karanovic, Gorana Nikolic, Jasmina Markovic-Lipkovski, Sanja Radojevic Skodric, Jelena Nesovic Ostojic

**Affiliations:** 1Institute of Pathology, Faculty of Medicine, University of Belgrade, 11000 Belgrade, Serbia; 2Clinic of Nephrology, Clinical Center of Serbia, Faculty of Medicine, University of Belgrade, 11000 Belgrade, Serbia; 3Institute of Pathological Physiology, Faculty of Medicine, University of Belgrade, 11000 Belgrade, Serbia; 4Institute for Medical Research, Department of Cardiovascular Physiology, National Institute of Republic of Serbia, University of Belgrade, 11000 Belgrade, Serbia

**Keywords:** PAX8, human kidneys, acute kidney injury, chronic kidney disease

## Abstract

Transcription factor PAX8, expressed during embryonic kidney development, has been previously detected in various kidney tumors. In order to investigate expression of PAX8 transcription factor in acute kidney injury (AKI) and chronic kidney diseases (CKD), immunohistochemical analysis was performed. Presence, location and extent of PAX8 expression were analyzed among 31 human kidney samples of AKI (25 autopsy cases, 5 kidney biopsies with unknown etiology and 1 AKI with confirmed myoglobin cast nephropathy), as well as in animals with induced postischemic AKI. Additionally, expression pattern was analyzed in 20 kidney biopsy samples of CKD. Our study demonstrates that various kidney diseases with chronic disease course that results in the formation of tubular atrophy and interstitial fibrosis, lead to PAX8 expression in the nuclei of proximal tubules. Furthermore, patients with PAX8 detected within the damaged proximal tubuli would be carefully monitored, since deterioration in kidney function was observed during follow-up. We also showed that myoglobin provoked acute kidney injury followed with large extent of renal damage, was associated with strong nuclear expression of PAX8 in proximal tubular cells. These results were supported and followed by data obtained in experimental model of induced postischemic acute kidney injury. Considering these findings, we can assume that PAX8 protein might be involved in regeneration process and recovery after acute kidney injury. Thus, accordingly, all investigation concerning PAX8 immunolabeling should be performed on biopsy samples of the living individuals.

## 1. Introduction

PAX8 is a member of the mammalian Paired-box family of genes that includes 9 (PAX1–9) different genes and plays a crucial role in the organogenesis of the kidney, thyroid gland, and Müllerian organs [1,2,3,4].

The reactivation of embryonic genes has been observed during pathogenesis of many diseases [5]. Sometimes, initial activation of such condition serves as a counteracting mechanism in order to diminish damaging signal and to improve wound healing [6,7]. However, the line between beneficial and adverse reactions could be loose, since constitutive activity of embryonic gene program during persistent alteration would also impair regeneration tissue efforts.

Among various embryonic genes expressed during kidney development, the fundamental role in branching morphogenesis and nephron differentiation belongs to PAX8 transcription factor [8]. Nuclear transcription factor PAX2 expressed by proliferating stem cells, give rise to renal epithelial cells. Once the epithelial cells differentiate into mature proximal and distal tubules, PAX2 expression ends, whereas expression of the related Pax8 protein continues [9]. Despite abundant evidence of its involvement in kidney development [8,10], only a few studies were focused on PAX8 expression in adult human kidneys emphasizing its significance in primary and metastatic renal cell carcinomas [11,12,13]. PAX8 positivity has also been reported in several cases of renal, thyroid, and ovarian tumors [11,14,15,16]. On the other hand, due to PAX8 high sensitivity and absence of expression in lung, breast, gastrointestinal, and mesothelial malignancies, PAX8 immunohistochemistry is now routinely used as an adjunctive tool in diagnosing tumors, especially in the cases when they become metastatic [17,18,19,20].

In contrast to animals, PAX8 is not completely down-regulated in normal adult human kidneys and remains expressed by some epithelial structures such as parietal cells of the Bowman’s capsule, distal tubules, loops of Henle and collecting ducts [11,12]. However, the presence of PAX8 in proximal tubular epithelial cells is not completely clarified. Tong and coworkers reported weak nonspecific cytoplasmic staining of PAX8 in proximal tubules in human adult kidneys [11], but the diagnostic applications of PAX2 have been studied more extensively than PAX8.

Since Ozcan et al. reported PAX8 nuclear expression in atrophic renal tubular epithelial cells regardless of nephronic segments [12,13], we considered an opportunity to investigate relationship of PAX8 nuclear positivity in the epithelial structures of the human adult kidney affected with various chronic diseases clinically presented with proteinuria, as well as to compare it with expression pattern in patients with acute kidney injury and to assess the clinical relevance of PAX8 detection in kidney biopsies. Moreover, appreciating an involvement of PAX8 in activation of BCL-2 transcription [21], as well as repression of p53 [22], we further wanted to explore expression of these genes in parallel with PAX8 immunolabeling.

## 2. Materials and Methods

### 2.1. Human Kidney Samples and Patients Data

The current study included 26 human kidney biopsy samples and 25 cadaveric kidneys. Their biopsies were submitted to routine diagnostic at Institute of Pathology, Faculty of Medicine, University of Belgrade during from 2012 to 2022, and the rest of tissue specimens were used for further research purposes. During routine work-up, biopsies were analyzed using optical microscopy, applying HE, PAS, Massone-trichrome stain, as well as using immunofluorescent analyzes with FITC conjugated antibodies: IgA, IgG, IgM, C1q, C3, fibrinogen, kappa and lambda. Cadaveric samples were obtained during routine autopsies performed at our Institute.

Kidney biopsies were obtained from 20 patients clinically presented with nephrotic syndrome with underlying different chronic kidney diseases, and from 6 patients who suffered from acute kidney injury (AKI).

Patients with chronic kidney diseases were diagnosed as follows: membranous glomerulonephritis (MGN, 6 patients), focal segmental glomerulosclerosis (FSGS, 3 patients), end-stage kidney disease (ESKD, 3), amyloidosis nephropathy (1 patient), diabetic nephropathy (1), malignant hypertension (1), benign hypertension (1), mesangioproliferative glomerulonephritis-IgM nephropathy (1), mesangioproliferative glomerulonephritis-IgA nephropathy (1), lupus nephritis class 4 (1) and membranoproliferative glomerulonephritis (1 patient).

Acute kidney injury samples were patients with AKI diagnosis established by biopsy with no other associated histopathological diagnosis of glomerulonephritis or glomerulopathy. All patients with AKI were with different duration and stage of the disease, and usually had clinical manifestations of nephrotic syndrome. According to the biopsy analysis, in 5 patients, the etiology of AKI has not been established, so they were lead as AKI of unknown etiology. One patient had a medical history of vomiting; the second had diarrhea; the third had no complaints except signs of nephrotic syndrome; the fourth was suffering from cardiomyopathy, and complained of loss of breath suddenly followed by edema. The fifth was presented with nephrotic syndrome with normal value of serum creatinine and urea, but after 6 months, there was a sudden increase in serum creatinine up to 700 µmol/L that was the reason why the biopsy was performed. The sixth patient was with established diagnosis of polymyositis with sudden increase in serum creatinine up to 248 µmol/L when the biopsy was performed and confirmed obstructive AKI due to mioglobinuria and abundant myoglobin casts. Additionally, 25 cadaveric kidneys of patients suffered from AKI induced by shock (hypovolemic and cardiac), whose autopsies were performed at our Institute, were also explored. We used medical records, to collect clinical and laboratory patients’ data recorded at the time of biopsy, as well as at the time of last medical examination. Patients were clinically classified in chronic kidney disease (CKD) stages, according to widely accepted recommendations [23]. We included the patients with CKD with different degree of tubulointerstitial lesions.

This part of the study was carried out in accordance with the Code of Ethics of the World Medical Association (Declaration of Helsinki) and was approved by the Ethic Committee of Medical Faculty University of Belgrade (approval no. 29/II-15).

### 2.2. Animals Samples with Induced Postischemic AKI

Since we had only 6 patients with AKI, we decided to performed experimental model of induced postischemic AKI in Wistar and spontaneously hypertensive (SHR) rats.

In these experiments, we used male spontaneously hypertensive (SHR, descendants of breeders originally obtained through Taconic Farms, Germantown, NY, USA) and normotensive Wistar rats 24 weeks old and about 300 g weights. The rats were bred at the Institute for Medical Research, University of Belgrade, Serbia, and housed under controlled laboratory conditions including constant temperature 22 ±   1 °C, humidity of 65  ±   1%, and 12 h light/dark cycle. These animals were kept in groups of three animals per cage and fed with a standard chow for laboratory rats (Veterinarski zavod, Subotica, Serbia). They had free access to food and water. All experimental animals were monitored at least once per day, throughout the course of the study.

The animals were randomly divided into three experimental groups: sham-operated Wistar rats (n = 3), Wistar rats with induced postischemic AKI (n = 3) and spontaneously hypertensive rats with induced postischemic AKI (n = 3).

AKI was induced by removal of the right kidney on previously anaesthetized rats (35 mg/kg b.m. sodium pentobarbital intraperitoneally) and atraumatic clamp occlusion of the left renal artery for 45 min. In the sham-operated group, the same surgical procedure was applied, but without left renal artery clamping. At the end of AKI induction, the wound abdominal incision was made and rats were returned into cages for 24 h, with free access to food and water. Ketoprofen (5 mg/kg b.m.) was administrated subcutaneously in order to alleviate postoperative pain. The animals were sacrificed by pentobarbital overdose injection. The kidney tissue was removed immediately after sacrificing and then prepared for histological examination.

For this experimental part of study, experimental protocol was carried out in accordance to the National Law on Animal Welfare (“Službeni Glasnik” no. 41/09) that follow the guidelines for animal research and principles of the European Convention for the Protection of Vertebrate Animals Used for Experimental and Other Purposes (Official Daily N. L 358/1-358/6, 18 December 1986) and Directive on the protection of animals used for scientific purposes (Directive 2010/63/EU of the European Parliament and of the Council, 22 September 2010), and was approved by the Ethic Committee of the Institute for Medical Research, University of Belgrade, Serbia (No. 323-0702569/2018-05/2).

### 2.3. Immunohistochemistry

After deparaffinization and rehydration, immunostaining was applied on 5 μm thick paraffin kidney sections treated for 20 min in low (pH 6.0) or in high pH (pH 9.0) buffer, depending on the antibody. Samples were incubated for 1 h at room temperature with primary PAX8 (RTU, MRQ-50, Cell Marque, Rocklin, CA, USA), CD10 (1:100, M7308, Dako, Glostrup, Denmark), nucleophosmin (1:200, MS-1849-P, Thermo Fisher, NeoMarkers, Fremont, CA, USA), BCL-2 (1:100, NCL-L-bcl2, Leica Biosystems, Novocastra, Newcastle, UK) and p53 (1:50, LCL-L-p53, Leica Biosystems, Novocastra, Newcastle, UK), and myoglobin (1:700, 9068-p. NeoMarkers, Fremont, CA, USA) antibodies. The EnVision^TM^ staining method (Dako, Glostrup, Denmark), visualization of antigen-antibody reaction by 3,3′-diaminobenzidine (DAB) and subsequent counterstaining with hemalaun (Merck, Burlington, MA, USA) were conducted to the majority of antibodies. However, visualization of antigen antibody reaction for myoglobin was performed with 3-amino-9-ethylcarbazole (AEC), instead of DAB. Negative controls were performed by omitting the first antibody. Slides were evaluated using the light microscope BX53 with DP12 CCD camera (Olympus, Hamburg, Germany).

Extension of interstitial renal fibrosis (IRF) was semi-quantitatively assessed on biopsies stained with PAS and Massone-trichrome, applying a scale from 0 to 3 with 0 meaning no IRF: 1—less than 25% of renal tissue with IRF; 2—25% to 50% of renal tissue with IRF; and 3—more than 50% of renal tissue with IRF. Using the same rule, the abundance of tubular atrophy (TA) was assessed (TA-0—no tubular atrophy; TA-1—less than 25% of atrophic tubuli; TA-2—25–50% of atrophic tubuli; TA-3—more than 50% of atrophic tubuli).

### 2.4. Statistical Analysis

Statistical analysis was performed using the IBM SPSS software. After applying the Chi-square, Fisher’s, Kruskal–Wallis, Mann–Whitney U, Student *t*-test and/or ANOVA, as well as Wilcoxon or t-test for paired samples, *p* values < 0.05 were considered to be significant. Graphs were made using Microsoft Office Excel software package.

## 3. Results

### 3.1. Immunohistochemical Detection of PAX8 in Human Kidney Biopsies and Autopsy Cases

Within 26 kidney biopsy samples, PAX8 was constitutively expressed in the nuclei of distal tubules (Figure 1A) and collecting ducts (Figure 1B), either cortical or medullar, in both cases of chronic and acute renal parenchymal damages. In addition, the majority of cases revealed also PAX8 nuclear positivity in the epithelium of loop of Henle (Figure 1B), as well as in the parietal cells of Bowman capsule (Figure 1A,C). However, these observations were not related to any pathohistological or clinical patients’ characteristics. Immunohistochemically, normal proximal tubular epithelial cells were negative for PAX8 nuclear staining (Figure 1A). Nevertheless, mostly in patients with nephrotic syndrome, atrophic renal tubular epithelial cells regardless of nephronic segments (Figure 1C–E), as well as non-atrophic proximal tubular epithelial cells close to the area of renal interstitial fibrosis (Figure 1D,E), expressed PAX8 in the nuclei.

On the other hand, various degree of tubular dilatation with flattened tubulocytes and interstitial edema, represented pathohistological signs of AKI (Figure 2A,B). In 5 kidney biopsies of AKI with unknown etiology only a few proximal tubular epithelial cells could be marked as weakly PAX8 positive (Figure 1F). Moreover, all 25 autopsy cases of acute kidney injury were devoid of PAX8 staining, even considering kidney structures such as collecting ducts and distal tubuli (Figure 1G) that were always positive in all examined biopsy samples (Figure 1A,B) and consequently used as internal positive control for PAX8 immunostaining. In order to clarify if the PAX8 specifically underwent post-mortal autolysis or the other antigens were also destroyed, we further performed CD10 (Figure 1H) and nucleophosmin (Figure 1I) immunolabeling. Positive membranous CD10 and nuclear nucleophosmin staining clearly showed post-mortally preserved antigens in paraffin embedded kidney tissues, indirectly indicated that PAX8 specifically underwent fast autolysis post-mortem.

Human kidney biopsy sample of patient with severe rabdomyolysis morphologically corresponded to myoglobin cast nephropathy with all features of AKI. Compared to all other biopsy samples of AKI, where PAX8 was induced in few proximal tubular epithelial cells, this case of myoglobin induced AKI was followed with widespread PAX8 expression in almost all tubular epithelial cells (Figure 3).

### 3.2. Immunohistochemical Detection of PAX8 in Kidney Animal Samples

Considering that strong conclusions about PAX8 relevance for AKI in humans could not be made due to insufficient number of kidney biopsies, we became encouraged to investigate PAX8 expression in animal models of AKI in order to perform a kind of translational research. Surprisingly, here, we found the same result as in the human biopsy case of AKI due to rhabdomylosis: the strong PAX8 expression was induced after induction of ischemic AKI, both in normotensive and hypertensive rats, with expression present also in proximal tubular epithelial cells (Figure 4B,C), while in control, sham operated group, weak PAX8 expression was noticed in rare tubular epithelial cells (Figure 4A).

### 3.3. Clinicopathological Relevance of PAX8 Expression in Proximal Tubuli

PAX8 nuclear expression in the proximal tubuli was significantly associated with interstitial fibrosis and tubular atrophy (*p* < 0.001), independently of the IRF and TA stages, Figure 5. Among twelve patients with interstitial fibrosis, four patients were in IRF stage 1, three patients were in IRF stage 2, and five patients were in advanced stage (IRF stage 3). The majority of them had PAX8 expression in proximal tubuli, whereas only one patient had initial phase of IRF (stage 1) and did not develop PAX8 expression in proximal tubuli. Moreover, all cases with tubular atrophy (three patients in TA stage 1, six patients in TA stage 2 and one patient in TA stage 3) had nuclear PAX8 expression in atrophic tubuli, as illustrated in Figure 5. Among 25 biopsy samples (five patients with unknown AKI etiology and twenty patients with nephrotic syndrome; patient with AKI due to rhabdomyolysis excluded from analysis due to young age), PAX8 nuclear expression in the proximal tubuli was observed in 11 cases, which revealed some degree of interstitial renal fibrosis (IRF), (Figure 5A), followed by tubular atrophy in 10 cases (Figure 5B). One patient had slight interstitial fibrosis (IRF-1) without tubular atrophy. In that case, proximal tubuli located close to the area of interstitial fibrosis also expressed PAX8 in the nuclei. Thus, nuclear detection of PAX8 in proximal tubuli was associated with chronic alterations of tubulointerstitial compartment. These observations were also supported by clinical and laboratory data of 20 patients suffering from nephrotic syndrome caused by different chronic kidney diseases. More increased serum creatinine levels, lower eGFR and even higher proteinuria levels were detected in patients with PAX8 nuclear detection in proximal tubuli, although these differences did not reach statistical significance (Table 1 and Table 2). However, we observed that patients with PAX8 in proximal tubuli had even higher serum creatinine (185.0 ± 136.3 μmol/L) and urea levels (17.2 ± 7.5 mmol/L) at the control medical examination (after median follow-up of 207 days, range: 44–279 days), compared to their values at the time of biopsy (serum creatinine: 167.2 ± 52.4 μmol/L; urea: 10.9 ± 2.8 mmol/L). The changes in urea values were statistically significant (*p* = 0.028).

The duration of symptoms and signs of kidney diseases prior biopsies were variable. Those with glomerulonephritis and glomerulopathies were admitted to hospital after median follow up of 240 (range: 21–6570) days of symptoms duration. Patients with AKI were admitted for biopsy after median follow up of 13 (range: 5–180) days of symptoms duration.

### 3.4. Relationship of PAX8 with BCL-2 and p53 Expressions

Expression of BCL-2 did not show any regularity. Among 25 kidney biopsies, used for determination of clinicopathological relevance of PAX8 expression in proximal tubuli kidney biopsies, it was detectable in morphologically preserved and destroyed kidney structures. In general, BCL-2 was not expressed abundantly like PAX8. Thus, parietal cells of Bowman’s capsule were sometimes BCL-2 positive. Scattered BCL-2 positivity was also observed among morphologically normal tubular cells either of proximal and distal tubuli or collecting ducts. Nevertheless, atrophic tubuli did not necessarily express BCL-2, showing huge heterogeneity among cases. Sometimes, none of the cells belonging to atrophic tubuli were BCL-2 positive, while occasionally all atrophic tubuli showed at least one BCL-2 positive tubular epithelial cell (Figure 6A). However, the most frequent pattern was the presence of rare BCL-2 positive cells within some of the atrophic tubuli.

In general, p53 was detectable only in few cases, staining primarily the nuclei of some tubular cells both normal and atrophic (Figure 6B). The same case is presented with PAX8 tubular immunopositivity, as shown on Figure 6C.

## 4. Discussion

In contrast to other organs [12,24,25,26,27,28], the involvement of PAX8 has not been widely studied in kidney diseases. A few studies examined its expression in renal tumors [12,13], but for the first time here we present PAX8 expression in non-tumor kidney diseases including those with chronic course, as well as cases of acute kidney injury. Additionally, we analyzed renal expression of PAX8 in experimental model of postischemic acute kidney injury in Wistar and spontaneously hypertensive (SHR) rats.

In this study, we observed expression of PAX8 in all atrophic tubuli regardless of the nephronic segment, as well as in the parietal cells of Bowman capsule in the cases of end stage kidney disease. Additionally, we found widespread interstitial fibrosis and tubular atrophy accompanied with PAX8 detection in both atrophic tubuli and non-atrophic proximal tubular cells close to the area of renal interstitial fibrosis. On the contrary, biopsy samples of the patients with acute kidney injury with unknown etiology showed only a few proximal tubular epithelial cells marked as weakly PAX8 positive. However, we reported strong nuclear PAX8 expression in proximal tubular cells in acute kidney injury provoked by myoglobin casts tubular obstruction. These results might be explained by different extent of renal damage in these cases. Actually, in patients with AKI of unknown etiology, the structure of renal tissue was almost normal and we can assume that the intensity of kidney damage was not enough stimulus to initiate PAX8 re-expression. On the other hand, in the case of obstructive AKI induced by myoglobin casts in tubular system, there are many structural changes confirming extensive tubular cells impairment that can provoke PAX8 expression. As the patients were in different stages and duration of AKI, we can say that the extent of the structural damage observed as a sign of severe renal injury may be more important stimulus for PAX8 expression then duration and stage of AKI itself. This assumption is in accordance with our results obtained in experimental model of induced postischemic AKI. We reported strong nuclear PAX8 expression in proximal tubular cells in all animals, Wistar and SHR with induced AKI. Postischemic AKI is followed with large extent of damage including necrosis and apoptosis of tubular cells, as well as the presence of many tubular casts. The structural and functional restoration of the kidney depends on a suptile balance of growth and transcription factors that direct gene expression [29]. The signaling pathways that are activated during this process often resemble those observed during kidney embryonic period. Tissue regeneration consists of dedifferentiation, proliferation, as well as transdifferentiation processes [30]. The proximal tubular cells of adult kidney are terminally differentiated and are out of cell cycle. According to the available evidence, in response to acute kidney injury (AKI) from ischemia or nephrotoxicity, the surviving epithelial cells of proximal tubule can become mitotically active and renew the damaged tubules [6,31,32]. In a fact, after injury, surviving cells undergo dedifferentiation, receiving progenitor cell characteristics [33]. These undiffentiation cells possess a higher proliferation rate than adult cells in healthy kidneys [34]. This process of regeneration must require changes in gene expression of surviving epithelial cells, which may involve the reactivation of genes controlling development and proliferation [35]. Up to now, it was reported that PAX2, transcription factor, expressed during kidney organogenesis regulates transition of mesenchymal cells to an epithelial phenotype [36], and re-expressed after injury [37]. In this study, we showed re-expression of PAX8, and can speculate that this transcription factor also might have important role in process of regeneration. Additionally, it seems to be a valuable issue to explore these transcriptional factors that are selectively expressed during embryogenesis and potentially re-expressed after tissue injury and thereby possibly modulate and improve the regeneration process [37].

Since we obtained only six biopsies of the aforementioned patients, we further performed PAX8 immunostaining on 25 autopsy cases of AKI induced by shock (hypovolemic and cardiac). Surprisingly, instead of getting more conclusive data, we found something new. In fact, all 25 autopsy cases were completely PAX8 negative including structures which could be used as internal positive control in biopsy specimens (i.e., distal tubuli and collecting duct). Although all 25 autopsy cases were morphologically well preserved on light microscopy, it makes quality of the samples doubtful. Thus, we decided to stain the samples with CD10, as membranous antigen that must be positive in proximal tubuli of well-preserved tissues [38], as well as nucleophosmin which stains the nuclei of many cells within the kidney [39]. The positivity of both CD10 and nucleophosmin was detected within all the analyzed autopsy cases. Since, at the same time PAX8 was absent, we thought it must be due to extremely fast post-mortal autolysis of PAX8 antigen in humans. If this fast post-mortal autolysis is confirmed in further studies, this finding would indicate that PAX8 cannot be used as marker in post-mortem diagnosis of PAX8 positive tumors and their metastasis.

It is well known that PAX8 plays an important role in proper development of many organs, including kidneys [5,40,41,42]. However, its involvement in diseases is also recognized and the roles in cell survival and proliferation become clearer [25,26,27]. Thus, PAX8 overexpression leads to increased proliferation rate of differentiated epithelial cells, while PAX8 silencing results in apoptosis through a p53-dependent pathway [26]. In tumor cells, PAX8 could be an important regulator of cell survival, through activation of BCL-2 anti-apoptotic gene transcription and also p53 downregulation [21,22]. Moreover, regulation of telomerase activity could be also a mechanism of PAX8 promoted cell survival [27]. Hence, PAX8 overexpression in tumor cells promotes aggressive tumor behavior. However, it is still not clear why PAX8 is constitutively expressed within mature segments of the nephron, such as Bowman capsule, loop of Henle, distal tubules and collecting ducts [12], while preserved proximal tubuli are usually devoid of PAX8 positivity. Nowadays, there is some evidence that PAX8 and PAX2 proteins regulate urea transporters and aquaporins to control urine concentration in the adult kidney [9].

Changes in the PAX8 expression pattern could be visible in kidneys with chronically affected tubulointerstitial compartment where all atrophic tubuli always express PAX8. It is tempting to speculate that the presence of PAX8 in atrophic tubuli serves as a counteracting mechanism in order to promote survival of damaged tubular cells. This mechanism could be rather adverse in long-term disease course, because supporting the survival of damaged tubular epithelium PAX8 could prevent tubular regeneration. This hypothesis could be supported with our findings that worsening of kidney function has been observed in patients with PAX8 proximal tubular expression, regarding increased serum creatinine and urea level after follow-up period. However, it has to be further approved in larger patient cohort including also a larger follow-up period.

At the end of the discussion, regarding about potential clinical impact of PAX8 expression in chronic kidney diseases and acute kidney injury, we can assume that PAX8 expression in AKI followed by abundant morphological changes in renal tissue may be one of the potential mechanisms activating to promote tubular cell survival and renal regeneration. In cases with CKD, PAX8 expression was associated with tubular atrophy and interstitial fibrosis, so this molecular change may indicate further irreversible damage followed by severe functional impairment. Where there is a link between PAX8 expression in acute kidney injury and greater risk of transitioning to chronic kidney damage is a question that awaits some future studies.

The fact that we observed only six patients with AKI and extended our investigation with animals samples could be a limitation of this study, but we also want to underline that the PAX8 expression pattern in human samples of myoglobin cast AKI was very similar to the PAX8 expression in the animal experimental model of postischemic acute kidney injury, so we can say that the results obtained from our research are comparable and promising in terms of determining renal PAX8 expression and might have prognostic significance in acute and chronic kidney diseases.

## 5. Conclusions

Our study demonstrates that various kidney diseases with a chronic course that results in the formation of tubular atrophy and interstitial fibrosis lead to PAX8 expression in the nuclei of proximal tubules. Furthermore, patients with PAX8 detected within the damaged proximal tubuli should be carefully monitored, since deterioration in kidney function was observed during follow-up. We also showed that myoglobin provoked acute kidney injury followed with large extent of renal damage was associated with strong nuclear expression of PAX8 in proximal tubular cells. These results were supported and followed by data obtained in experimental model of induced postischemic acute kidney injury. Considering these finding, we can assume that the PAX8 protein might be involved in the regeneration process and recovery after acute renal injury. Thus, accordingly all investigation concerning PAX8 immunolabeling should be performed on biopsy samples of the living individuals.

## Figures and Tables

**Figure 1 diagnostics-12-02036-f001:**
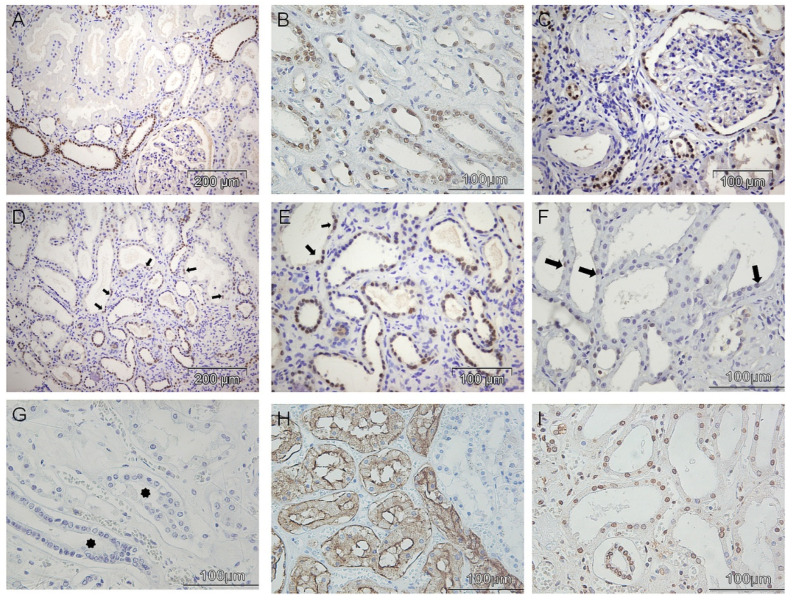
Immunohistochemical detection of PAX8 in kidney biopsies and autopsy cases. (**A**) Expression of PAX8 in the nuclei of distal tubules and in the parietal cells of Bowman capsule, as well as in the nuclei of few atrophic tubuli. (**B**) Collecting ducts and loop of Henle showed PAX8 nuclear positivity in the renal medulla. (**C**) The case of end stage kidney disease (ESKD) with PAX8 positivity in all the atrophic tubuli regardless of the nephronic segment, as well as in the parietal cells of Bowman capsule. (**D**,**E**) Widespread interstitial fibrosis and tubular atrophy; PAX8 detection in atrophic tubuli, as well as non-atrophic proximal tubular epithelial cells close to the area of renal interstitial fibrosis (arrows). (**F**) Biopsy sample of the patient with acute kidney injury (AKI) showed only a few proximal tubular epithelial cells marked as weakly PAX8 positive (arrows). (**G**) Autopsy case of AKI completely devoid of PAX8 staining, even considering kidney structures such as collecting ducts and distal tubuli (asterisks). (**H**) The same autopsy case of AKI showed CD10 membranous immunopositivity. (**I**) Preserved nucleophosmin expression in the same autopsy AKI case.

**Figure 2 diagnostics-12-02036-f002:**
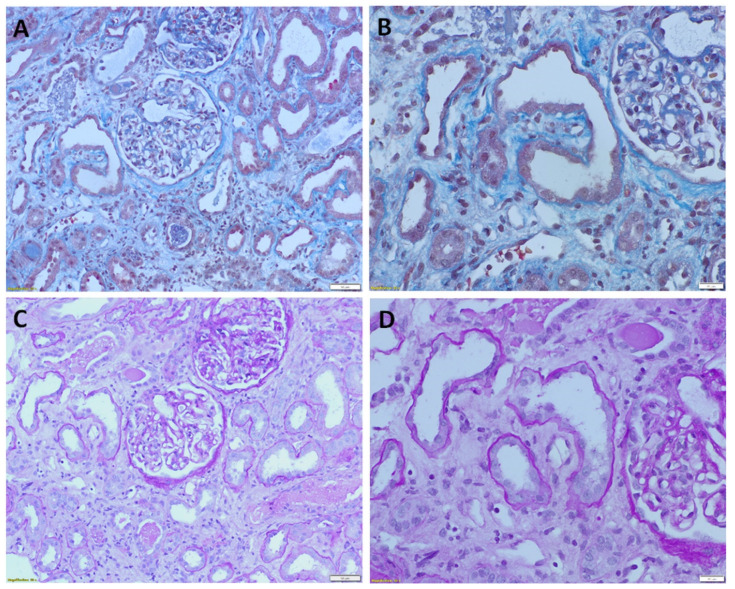
Morphology of acute kidney injury in human biopsy samples. Dilatation of proximal tubuli with flattened epithelium which are separated with widened interstitium due to edema. (**A**) Massone-trichrome stain, ×200. (**B**) Massone-trichrome stain, ×400. (**C**) PASS stain, ×200. (**D**) PASS stain, ×400.

**Figure 3 diagnostics-12-02036-f003:**
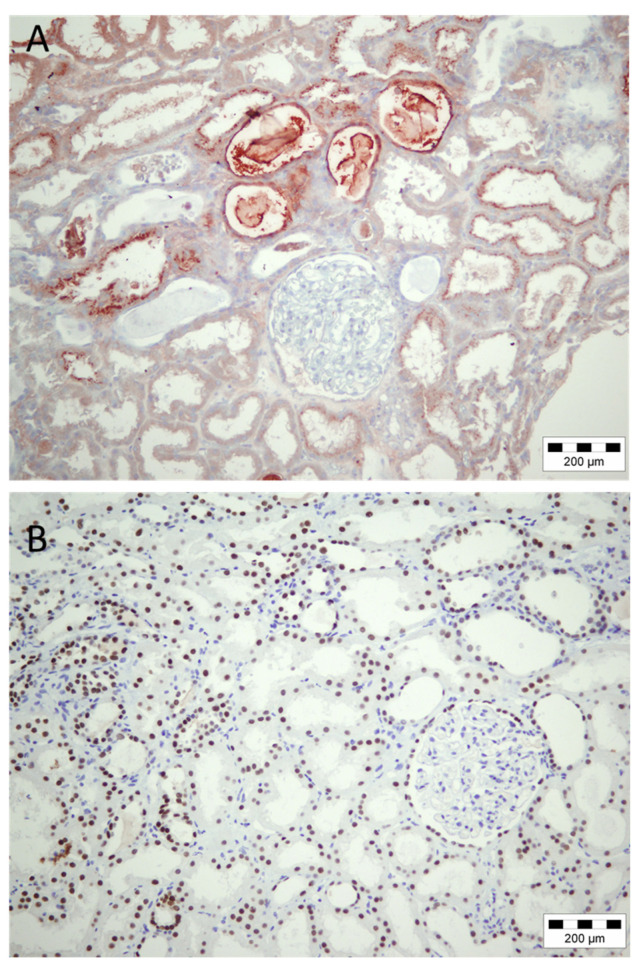
Acute kidney injury due to rhabdomyolysis. (**A**) Immunohistochemical staining with myoglobin revealed positive casts in luminal area of tubuli, as well as presence of myoglobin within some tubular epithelial cells. (**B**) Widespread immunohistochemical expression of PAX8 in myoglobin provoked acute kidney injury.

**Figure 4 diagnostics-12-02036-f004:**
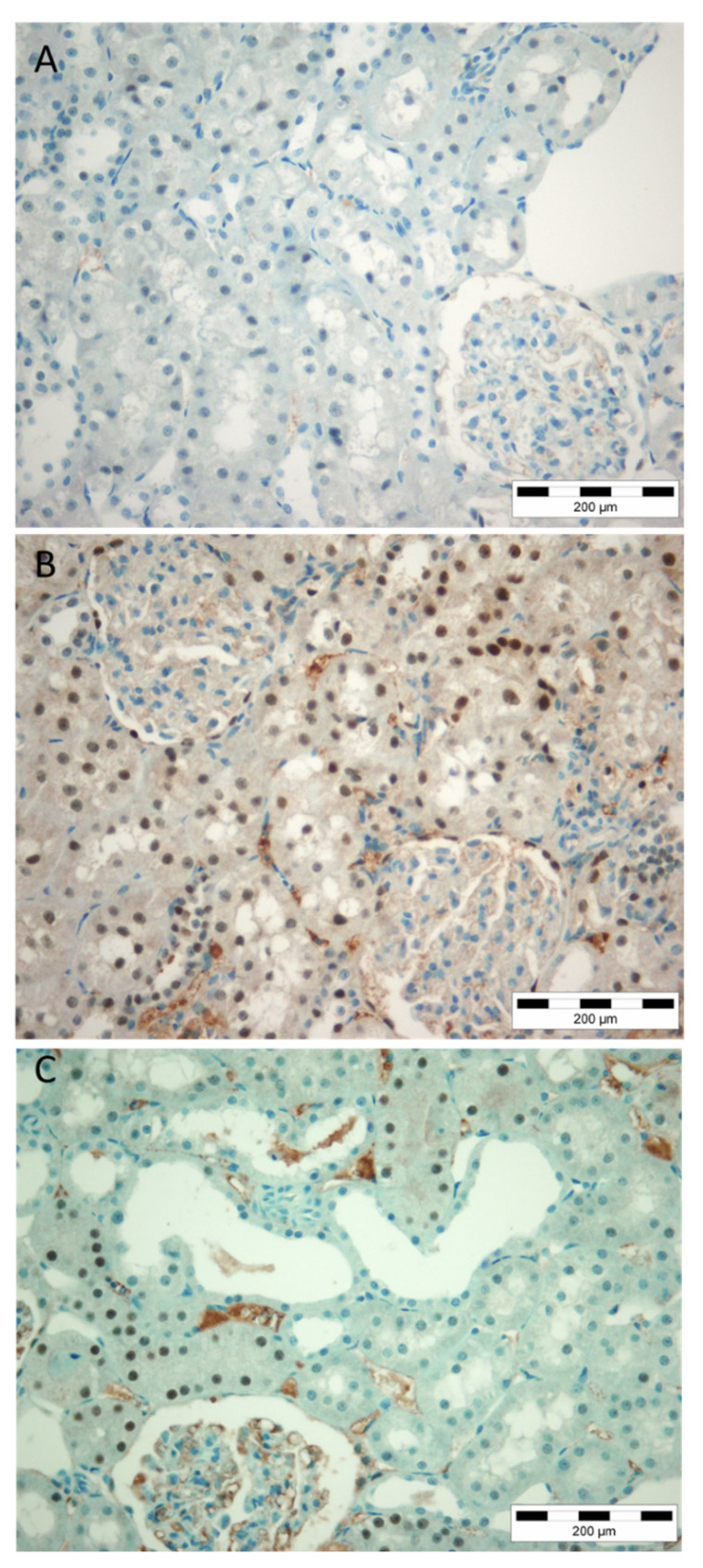
Immunohistochemical expression of PAX8 in rat kidneys with induced postischemic acute kidney injury. (**A**) Rare tubular epithelial cells with weak expression of PAX8 in control (sham operated) kidney. (**B**,**C**) Induction of PAX8 expression in proximal tubular epithelial cells in normotensive Wistar and spontaneously hypertensive rats after AKI induction.

**Figure 5 diagnostics-12-02036-f005:**
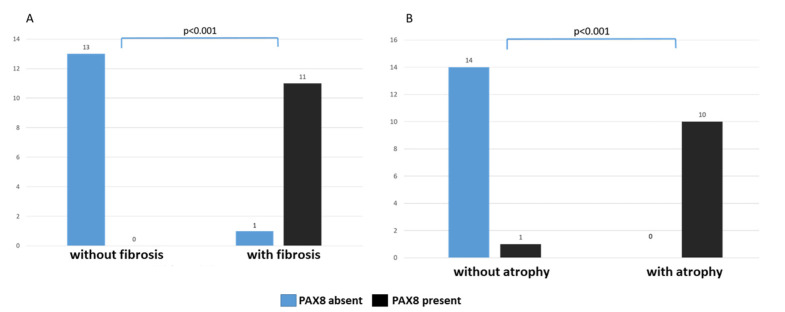
Relationship of PAX8 nuclear expression in the proximal tubuli with interstitial fibrosis and tubular atrophy. (**A**) Distribution of the patients according to positive PAX8 expression in proximal tubuli and stratified by the presence or absence of interstitial renal fibrosis (IRF). (**B**) Distribution of the patients according to the positive PAX8 expression in proximal tubuli and stratified by the presence and absence of tubular atrophy (TA).

**Figure 6 diagnostics-12-02036-f006:**
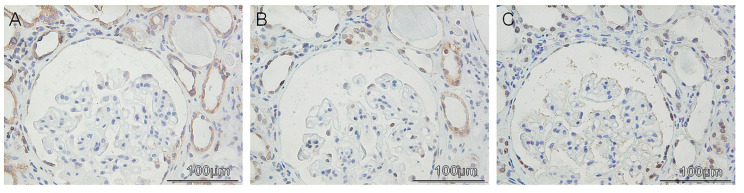
Immunohistohemical detection of BCL-2, p53 and PAX8 kidney biopsies. (**A**). BCL-2; (**B**). p53; (**C**). PAX8.

**Table 1 diagnostics-12-02036-t001:** Clinical and laboratory parameters of 20 patients clinically presented with nephrotic syndrome.

Clinical and Laboratory Parameters		At the Time of Kidney Biopsy			At the Time of Control Medical Examination		
		*PAX8 in Proximal Tubuli*		*p*	*PAX8 in Proximal Tubuli*		*p*
		*Absent*	*Present*		*Absent*	*Present*	
Age [year]	mean ± SD	35.1 ± 16.7	44.1 ± 18.1	0.243	39.0 ± 9.5	46.5 ± 22.1	0.601
Serum creatinine [μmol/L]		159.4 ± 166.1	167.2 ± 52.4	0.895	129.8 ± 52.2	185.0 ± 136.3	0.532
eGFR [mL/min/1.73 m^2^]		64.2 ± 36.9	42.8 ± 21.1	0.141	65.0 ± 44.2	46.5 ± 27.8	0.458
Urea [mmol/L]		11.4 ± 9.7	10.9 ± 2.8	0.886	9.8 ± 2.9	17.2 ± 7.5	0.154
Glucose [mmol/L]		5.3 ± 0.7	5.1 ± 1.3	0.247	3.9 ± 0.2	5.1 ± 0.9	0.094
Proteinuria [g/24 h]		5.9 ± 4.4	6.82 ± 4.8	0.673	3.7 ± 5.5	4.6 ± 5.4	0.794

eGFR—estimated glomerular filtration rate; Median follow-up was 207 days (range: 44–279 days).

**Table 2 diagnostics-12-02036-t002:** PAX8 expression related to CKD stage in 20 patients clinically presented with nephrotic syndrome.

CKD Stage	PAX8 in Proximal Tubuli	
	*Positive*	*Negative*
1	1	5
2	0	4
3A	1	0
3B	3	2
4	4	0
5	0	0

CKD stage was determined by calculating estimated glomerular filtration rate (eGFR) and following the widely accepted recommendations [23].

## Data Availability

The data presented in this study are available in article.

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
