# Peer review of "Clinicopathological Relevance of PAX8 Expression Patterns in Acute Kidney Injury and Chronic Kidney Diseases"

_diagnostics, 2022, doi:10.3390/diagnostics12092036_

Round 1
Reviewer 1 Report
Dear Authors,
I read the article carefully:
Clinicopathological relevance of PAX8 expression patterns in acute kidney injury and chronic kidney diseases.
Your work is interesting, but I recommend the following:
1. In subchapter 2.1. Human kidney samples and patients data, please specify during which period the study was conducted.
2. Please enter more details / a statistically clearer description for Figure 4.
3. The discussion chapter can be improved, please introduce more studies in the field and compare / complete them with your results. Also, please enter in the discussion chapter more details about the clinical impact on patients of these molecular changes.
4. Please correct typing and typing errors in English.
5. Please double-check the references
BR,
Author Response
Response to Reviewer 1 Comments
Point 1: 1. In subchapter 2.1. Human kidney samples and patients data, please specify during which period the study was conducted.
Response 1: Thank the reviewer on very useful comments and suggestions. Medical base of patients kidney biopsies from 2012. to 2022. was used. We included the patients with CKD with different degree of tubulointerstitial lesions. They were sufferig from different diagnosis, including membranous glomerulonephritis (MGN, 6 patients), focal segmental glomerulosclerosis (FSGS, 3 patients), end-stage kidney disease (ESKD, 3), amyloidosis nephropathy (1 patient), diabetic nephropathy (1), malignant hypertension (1), benign hypertension (1), mesangioproliferative glomerulonephritis - IgM nephropathy (1), mesangioproliferative glomerulonephritis - IgA nephropathy (1), lupus nephritis class 4 (1), and membranoproliferative glomerulonephritis (1 patient). Also, Acute kidney injury samples were patients with AKI diagnosis established by biopsy with no other associated histopathological diagnosis of glomerulonephritis or glomerulopathy. We incorporated these new information in subchapter 2.1. and marked up using the “Track Changes”, (see lines 83-85; 93-99; 117-120 in revised manuscript).
Point 2: Please enter more details / a statistically clearer description for Figure 4.
Response 2: We changed Figure 4. (now it is Figure 5, see lines 390-395) and added more details, (see lines 357-366). Hope new Figure, and its description is clearer now.
Point 3: The discussion chapter can be improved, please introduce more studies in the field and compare / complete them with your results. Also, please enter in the discussion chapter more details about the clinical impact on patients of these molecular changes.
Response 3: PAX8 and PAX2 involvment was widely investigated in kidney development, but only a few studies were focused on PAX8 and its expression in adult human kidneys emphasizing its role in primary and metastatic renal cell carcinomas. According to our knowledge, studies that followed PAX8 expression in non-tumor diseases are very rare (for example Ozcan et al. 2011, investigated PAX8 expression in non-neoplastic tissues, primary tumors and metastatic tumors), and here we presented for the first time PAX8 expression in chronic kidney disease as well as acute kidney injury. According to your suggestions, we added new parts in the discussion related to the potential clinical impact of these molecular changes on patients (see lines 575-591). Hope these added parts will improve discusssion.
Point 4: Please correct typing and typing errors in English.
Response 4: We corrected typing errors in English (see lines 53, 55, 60, 70, 153-154, 529, 531).
Point 5: Please double-check the references
Response 5: We double-checked the references and changed the references 23 (see lines 686-687) and 34 (line 713), and excluded the reference 39 (lines 723-725).
Sincerely,
Jelena Nesovic Ostojic
Reviewer 2 Report
This is an very interesting study of PAX8 expression patterns in acute kidney injury and chronic kidney diseases. I have only few comments
Comments
1. Including in the same paper both human samples of AKI, autopsy cases, CKD and animal samples may increase the number of samples however weaken the reported results. Please discuss this issue or report it as a limitations of the study
2. To expose better the clinical significance of your findings please report or analyse if there was any association between PAX8 expression and stage or duration of AKI
3. In 5 cases with unknown AKI please report histopathology findings, mainly those indicating tubular damage
4.Have the patients with AKI been examined for tubular proteinuria (ex. b2-microglobulin?) or other tubular function tests?
5. Patients with CKD are reported as proteinuric patients. However, there are no date on stage of the disease, cause of CKD, duration on CKD. Please provide data or discuss it.
Author Response
Response to Reviewer 2 Comments
Point 1: Including in the same paper both human samples of AKI, autopsy cases, CKD and animal samples may increase the number of samples however weaken the reported results. Please discuss this issue or report it as a limitations of the study
Response 1: Thank the reviewer on very useful comments and suggestions. We stated in the manuscript that we had only 6 patients with AKI and that we extended the number of samples by setting up an experimental model of induced postischemic acute kidney injury. According to your comment, we emphasized in the text that this can be the limitation of the study, but we also discussed that the results were comperable as PAX8 expression pattern in human renal samples was very similar to PAX8 expression in animals kidneys. We incorporated this new text in chapter Discussion and marked up using the “Track Changes”, (see lines 592-598 in revised manuscript).
Point 2: To expose better the clinical significance of your findings please report or analyse if there was any association between PAX8 expression and stage or duration of AKI
Response 2: Six patients with AKI were with different duration and stage of the disease, and mostly had clinical manifestations of nephrotic syndrome. One patient had medical history of vomiting, the second was with diarrhea, third denied complaints except signs of nephrotic syndrome, fourth was suffering from cardiomyopathya, complained on loss of breath suddenly followed by edema. Fifth was presented with nephrotic syndrome with normal value of serum creatinine and urea, but after 6 months there was a sudden increase in serum creatinine up to 700 µmol/L that was the reason why the biopsy was peformed. Sixth patient was with established diagnosis of polimyositis with sudden increase in serum creatinine up to 248 µmol/L when the biopsy was performed. Analyzing renal samples of these patients, we can assume not duration nor stage of AKI was related to PAX8 expression, but more important was the abundance of renal damage. We incorporeted this in the text (see lines 117-120; 122-130; 501-503).
Point 3: In 5 cases with unknown AKI please report histopathology findings, mainly those indicating tubular damage
Response 3: We added a new figure (now it is Figure 2), that report histopathological findings in cases with unknown AKI (see lines 309-312).
Point 4: Have the patients with AKI been examined for tubular proteinuria (ex. b2-microglobulin?) or other tubular function tests?
Response 4: These patients were not tested for tubular proteinuria nor other tubular test were perform before biopsy. Maybe, these tests were done after biopsy, but we don’t have that information, and can’t guarantee it.
Point 5: Patients with CKD are reported as proteinuric patients. However, there are no date on stage of the disease, cause of CKD, duration on CKD. Please provide data or discuss it.
Response 5: We added causes of CKD of our patients in the subchapter 2.1 (see lines 93-99). Also, there is a new table (Table 2) with the information of the stage CKD (see lines 455-457) and we added duration of CKD (see lines 385-389).
Sincerely,
Jelena Nesovic Ostojic
Round 2
Reviewer 2 Report
I have no further comments
thank you